# HMGA2 Overexpression in Papillary Thyroid Cancer Promotes Thyroid Cell Dedifferentiation and Invasion, and These Effects Are Counteracted by Suramin

**DOI:** 10.3390/ijms26041643

**Published:** 2025-02-14

**Authors:** Cindy Van Branteghem, Nicolas Henry, Ligia Craciun, Carine Maenhaut

**Affiliations:** 1IRIBHM—Jacques E. Dumont, Université Libre de Bruxelles, 1070 Brussels, Belgium; cindy.van.branteghem@ulb.be (C.V.B.); nicolas.a.henry@ulb.be (N.H.); 2Anatomie Pathologique, Hôpital Universitaire de Bruxelles, Université Libre de Bruxelles, 1070 Brussels, Belgium; lcraciun@cerbaresearch.com

**Keywords:** HMGA2, papillary thyroid cancer, MAPK, dedifferentiation, forskolin, suramin

## Abstract

Thyroid cancer is the most prevalent endocrine malignancy, and papillary thyroid carcinoma (PTC) is the most common type of thyroid malignancy. While PTC generally has a favorable prognosis, a subset dedifferentiates into aggressive forms. However, the molecular mechanisms responsible for aggressiveness and dedifferentiation are still poorly understood. We previously showed that HMGA2, a non-histone architectural transcription factor overexpressed in PTC, is involved in cell invasion. This study aimed to further analyze the role of HMGA2 in PTC tumorigenesis by exploring the expression of thyroid-specific and EMT-related genes following HMGA2 knockdown in thyroid cancer cell lines. Then, the clinical relevance of our data was evaluated in vivo. HMGA2 silencing did not modulate the expression of EMT related genes but led to the increased expression of thyroid differentiation genes. Our data also suggest that the MAPK pathway induces thyroid cell dedifferentiation through HMGA2. On the other hand, forskolin, promoting thyroid differentiation, decreased HMGA2 expression. The negative correlations between HMGA2 and thyroid-specific gene expressions were confirmed in a transgenic mouse model of PTC and in human PTC. Finally, we showed that HMGA2 inhibition by suramin reduced cell invasion and induced differentiation expression in vitro, indicating a new therapeutic strategy for treating thyroid cancer.

## 1. Introduction

Thyroid cancer (TC) is identified as the most prevalent malignant tumor within the endocrine system, with a rising global incidence rate [1]. Papillary thyroid carcinomas (PTC) constitute the most common type of thyroid malignancies (80–85%) and are primarily caused by genetic alterations in BRAF and RAS genes, along with RET/PTC rearrangements, resulting in the constitutive activation of the mitogen-activated protein kinase (MAPK) signaling pathway. This pathway is well recognized for inducing proliferation and dedifferentiation of thyrocytes [2]. While PTC generally has a favorable prognosis with conventional therapeutic interventions like thyroidectomy and radioactive iodine (RAI) therapy, a subset (10–15%) presents hallmarks of local invasion, recurrences and distant metastases. Two-thirds of these cases manifest a loss of iodine-131 uptake, known as RAI-refractory PTC, which results from the dedifferentiation of thyroid follicle cells through the abnormal silencing of both thyroid-specific genes and transcription factors. The development of RAI-refractory PTC, which remains a therapeutic challenge, is predominantly linked to genetic and epigenetic alterations, along with dysregulated signaling pathways [3]. In response to this challenge, various compounds have been investigated in preclinical and clinical studies with the aim to restore RAI uptake. However, conclusive trials in this regard are still pending. Thus, a deeper understanding of the molecular mechanisms driving the dedifferentiation process leading to RAI refractoriness may pave the way for the development of novel therapeutic redifferentiation strategies.

HMGA2 (high mobility group A2) is a non-histone architectural transcription factor which binds to AT-rich sites in the minor groove of DNA, thereby contributing to transcriptional regulation of numerous genes by altering chromatin structure. During embryogenesis, HMGA2 is highly expressed, playing a pivotal role in maintaining stemness and regulating the differentiation process [4,5]. However, it becomes an oncoprotein when expressed in adult cells [4,6]. In thyroid cancer, HMGA2 is overexpressed and is already a well-described molecular marker able to distinguish between benign and malignant thyroid tumors [7,8,9,10]. In our previous study describing the role of miR-204-5p and HMGA2 in PTC, we proposed that inhibiting HMGA2 could offer promising perspectives for thyroid cancer treatment. Indeed, we demonstrated the involvement of HMGA2 in cell invasion and its direct regulation by miR-204-5p, a microRNA identified in papillary thyroid carcinoma as downregulated and inversely associated with aggressiveness [11,12]. In addition, accumulating evidence supports a role for HMGA2 in epithelial-mesenchymal transition (EMT) in various cancers including pancreatic, prostate and colorectal cancer [13,14,15], and its function as a modulator of the TGFβ signaling pathway [16]. EMT is associated with partial dedifferentiation, resulting in cells with increased stemness. However, the interplay between HMGA2, dedifferentiation and EMT, and the TGFβ signaling pathway remains unexplored in the context of thyroid cancer. On the other hand, suramin, an antiparasitic drug known for treating African sleeping sickness, was recently discovered to be a potent inhibitor of HMGA2-DNA interactions. Despite this discovery, no studies evaluated its potential as therapeutic agent for thyroid cancer treatment.

This study aimed to improve our understanding of the role of HMGA2 in PTC tumorigenesis by exploring the expression of thyroid-specific and EMT-related genes following HMGA2 knockdown in two thyroid cancer cell lines: TPC-1 and BCPAP cells. Our findings show than HMGA2 does not impact the EMT process at transcriptional level and does not modulate it through the TGFβ signaling pathway. We uncovered a crucial role for HMGA2 in inhibiting thyroid differentiation induced by the MAPK signaling pathway, by repressing the expression of thyroglobulin (TG), thyroperoxidase (TPO) and PAX8. The clinical relevance of our results and the negative correlations between HMGA2 and thyroid specific gene expressions were analyzed in a mouse model of PTC, the RET/PTC3 transgenic mice [17,18], as well as in human PTC. Data was gained by analyzing RNAseq data from TCGA and independent PTC samples. Finally, we explored in the TPC-1 and BCPAP cell lines the therapeutic potential of suramin for treating thyroid cancer. Our data revealed that administration of suramin significantly inhibited invasion and dedifferentiation of TPC-1 and BCPAP cells, indicating a promising new therapeutic strategy for treating thyroid cancer.

## 2. Results

### 2.1. HMGA2 Silencing Does Not Modulate the Expression of EMT Related Genes in PTC Cell Lines

In our previous study, we showed that HMGA2 silencing by siRNA as well as MAPK signaling inhibition decreased invasion in TPC-1 and BCPAP cells, while did not impact proliferation or apoptosis [11]. We also showed that the MAPK signaling pathway regulates EMT-related genes and HMGA2 expression in PTC cell lines. Since HMGA2 contributes to the transcriptional regulation of numerous genes by binding to DNA, we hypothesized that the MAPK signaling pathway may govern the expression of EMT-related genes through HMGA2. To investigate this, we performed RT-qPCR analyses on the main EMT-related genes, three days after specifically knocking down HMGA2 using siRNA transfection in TPC-1 and BCPAP cells, both having high levels of endogenous HMGA2 expression [11]. To minimize non-specific effects due to off-target gene silencing, commonly present when transfecting siRNA at high concentrations (>100 nM), we used a 20 nM concentration of siRNA for the transfections, lower than those used in most similar studies [19,20]. To verify that the control itself did not have any effects, we also included a group of non-transfected (NT) cells to monitor the effects of transfection, obtain basal gene expression and detect any non-specific effects of the siRNA-negative control. HMGA2 siRNA transfection effectively reduced both HMGA2 mRNA and protein expression 72 h post-transfection, as described in our previous study [11]. For both cell lines, no significant changes in the mRNA expression levels of SNAI1, SNAI2, ZEB1, CDH1 and SOX4 compared to cells transfected with a control siRNA was observed (Figure 1), suggesting that HMGA2 does not modulate the transcription of those EMT-related genes.

### 2.2. HMGA2 Silencing Does Not Modify the Expression of Genes Triggering EMT Following TGFβ Treatment in PTC Cell Lines

Since several studies characterized HMGA2 as an inducer of EMT in response to TGFβ through the regulation of SNAI1/2 expression [21], we aimed to define the impact of HMGA2 on EMT in response to TGFβ in thyroid cells. We determined the optimal concentration and duration of TGFβ treatment required to induce the expression of genes known to be activated by this pathway and linked to the EMT process. We also analyzed by Western Blot the expression level of SMAD2 phosphorylation (P-SMAD2) following 30 min, 1 and 24 h of treatment with TGFβ1 at different concentrations (1, 10, 20 and 100 ng/mL) in TPC-1 cells. As shown in Appendix A, the TGFβ signaling pathway is not endogenously activated in TPC-1 cells, as evidenced by the absence of SMAD2 phosphorylation in cells not treated with TGFβ (C-). In contrast, 30 min of treatment with 1, 10, 20 or 100 ng/mL TGFβ1 led to the activation of the canonical TGFβ signaling pathway, characterized by the presence of SMAD2 phosphorylation (P-SMAD2). This activation persisted at least up to 72 h following a 10 ng/mL treatment (Appendix A). We further analyzed by RT-qPCR the expression of EMT-related genes in TPC-1 and BCPAP cells 72 h after transfection with a siRNA targeting HMGA2 or a control siRNA, as well as in non-transfected cells, all treated or not for 24 h with TGFβ1 (10 ng/mL). The experimental scheme is shown in Appendix A. As expected, treatment with TGFβ for 24 h of non-transfected TPC-1 and BCPAP cells induced the transcription of EMT-related genes such as SOX4, SNAI1 and SNAI2 and repressed TGFBR2 mRNA expression (Figure 2a) [22,23]. However, no significant difference in the mRNA expression of those genes was measured in HMGA2 siRNA transfected cells after stimulation with TGFβ (Figure 2a), suggesting that HMGA2 is not involved in their induction by TGFβ in these conditions. Additionally, we measured by RT-qPCR HMGA2 mRNA expression following 1, 20 or 24 h of TGFβ1 treatment in TPC-1 cells and observed no change in expression until 24 h of treatment while as expected, the mRNA levels of SNAI2, known to be induced by this pathway (see Appendix A) strongly increased (Figure 2b). These findings suggested that while TGFβ induced EMT-related genes transcription in thyroid cancer cells, HMGA2 did not influence this process. Additionally, HMGA2 mRNA expression did not appear to be regulated by TGFβ in these cells.

### 2.3. HMGA2 Silencing Induces the Expression of Thyroid Differentiation Genes in PTC Cell Lines

PTC progression is accompanied by dedifferentiation and constitutive activation of the MAPK signaling pathway which downregulates the expression of genes involved in iodine metabolism, dampening responses to RAI therapy [24]. Loss of differentiation following repression of thyroid-specific genes such as sodium/iodide symporter (NIS), thyroglobulin (TG), TSH receptor (TSHR), thyroperoxidase (TPO), transcription factors paired box gene 8 (PAX8) and thyroid transcription factor-1 (TTF1) is common. To investigate whether HMGA2 is involved in the dedifferentiation process, we examined the mRNA expression levels of TG, TPO and NIS as well as the transcription factor PAX8 in the TPC-1 and BCPAP cell lines. Cells were either transfected with a siRNA targeting HMGA2, with a control siRNA, or left untransfected. For both cell lines, a significant increase in the expression of TG, TPO and PAX8 mRNAs was observed after HMGA2 knockdown, as shown in Figure 3a. Basal expression of NIS mRNA was undetectable in both TPC-1 and BCPAP cell lines, preventing the interpretability of the findings regarding the effect of HMGA2 on NIS expression. Moreover, treatment of TPC-1 and BCPAP cells for 24 h with trametinib, a MEK inhibitor which was shown to reduce HMGA2 mRNA expression levels [11], increased TG, TPO and PAX8 mRNA levels (Figure 3b), suggesting that the MAPK pathway induces cell dedifferentiation in papillary thyroid carcinomas through HMGA2. Since thyroid cell differentiation is induced by the cAMP-dependent pathway, we treated the cells for 48 h with forskolin, an adenylate cyclase activator, to activate this pathway. In this condition, HMGA2 mRNA expression was diminished while TG, TPO and PAX8 mRNA levels were enhanced (Figure 3c), reinforcing the potential role of HMGA2 in thyroid dedifferentiation.

### 2.4. HMGA2 Is Overexpressed, While Thyroid Differentiation Genes Are Downregulated in the Thyroids of 2- and 6-Month-Old RET/PTC3 Mice

Cell lines serve as only partial models of tumors in vivo, yet they often inadequately represent the complexity of tumors. Among thyroid cell lines, the expression of differentiation genes is typically low, and in some cases, undetectable. To address the limitations of cell line models and investigate the in vivo relevance of our findings, we used the RET/PTC3 transgenic mouse model. These mice express in their thyroid and under the control of the thyroid-specific thyroglobulin promoter, the human RET/PTC3 rearrangement. The abnormal expression of the RET/PTC3 oncoprotein leads to the development of solid variant of PTC and constitutive activation of the MAPK and PI3K signaling pathways [17,18]. These mice have been extensively characterized in our laboratory, and are a good model of human PTC, especially in young animals (2 months) [18,25]. We have previously shown that HMGA2 expression was increased in the RET/PTC3 thyroids of 2- and 6-month-old mice [11]. Here, we measured the expression of thyroid differentiation markers and transcription factors by RT-qPCR. As shown in Figure 4a, at 2 months, HMGA2 mRNA expression was significantly increased, while NIS and TG mRNA levels were significantly decreased in the thyroids of RET/PTC3 mice compared to wild-type thyroids. On the other hand, the expression levels of TPO, TSHR, NKX2.1 and PAX8 remained similar between conditions. A significant inverse correlation between the mRNA expression of HMGA2 and that of NIS or TG was observed (R = −0.48, *p* = 0.04; R = −0.53, *p* = 0.016 respectively) (Figure 4b). In 6-month-old RET/PTC3 mice, which are in a more advanced stage of tumorigenesis, a similar increase in HMGA2 mRNA expression and decrease in NIS and TG mRNA expression were detected. In addition, a decrease in PAX8 mRNA expression was also present (Figure 4a). Again, an inverse correlation between the mRNA expression of HMGA2 and that of NIS, TG or PAX8 was observed (Figure 4b) (R = −0.45, *p* = 0.06; R = −0.57, *p* = 0.01, R = −0.81, *p* = 0.002 respectively). TPO mRNA was not modulated while TSHR and NKX2.1 mRNA levels were increased in the RET/PTC3 thyroids. The increased expression of HMGA2 and decreased expression of TG were confirmed at protein levels by immunostaining in the thyroids of 2- and 6-month-old RET/PTC3 mice (Figure 4c).

Our findings indicate that HMGA2 may be involved in the process of thyroid cell dedifferentiation by inhibiting the expression of some key thyroid differentiation genes and transcription factors, such as NIS, TG and PAX8, in accordance with our in vitro results.

### 2.5. HMGA2 mRNA Expression Is Inversely Correlated to TG, TPO, NIS and PAX8 mRNA Expression in Human PTC

Our group and others have already reported that HMGA2 expression is increased in PTC [11,26,27]. To assess the clinical relevance of our in vitro results, a correlation analysis between the mRNA expression level of HMGA2 and that of TG, TPO, NIS or PAX8 was undertaken, first by using the RNAseq data from the TCGA (The Cancer Genome Atlas) database. An inverse correlation was observed between the mRNA levels of HMGA2 and TG, TPO, NIS or PAX8 (R = −0,43, *p* = 2.028 × 10^−27^; R = −0,63, *p* = 2.624 × 10^−65^; R = −0.50, *p* = 1.094 × 10^−36^; R = −0.43, *p*= 7.332 × 10^−28^ respectively) as shown in Figure 5a. We further investigated their expression by RT-qPCR in an independent cohort of 11 PTC. Similar trends were observed, although not statistically significant due to the small number of samples, except for TPO for which the negative correlation with HMGA2 was significant (Figure 5b).

### 2.6. Immunohistological Analyses Reveal Different Thyroid Differentiation States in PTC with HMGA2+/TG− and HMGA2−/TG+ Cells

Our mRNA and correlation analyses were performed on bulk thyroid tissues. To investigate the relationship between HMGA2 and thyroid differentiation markers at cellular resolution, immunohistochemical analyses were undertaken. HMGA2 and TG were detected by immunofluorescence performed on two adjacent sections of different PTC. We confirmed the already reported overexpression of HMGA2 and underexpression of TG in PTC [11,26,27]. Interestingly, we noticed different thyrocyte differentiation states, with areas where HMGA2 was highly expressed (in yellow), while thyroglobulin (in green) was almost undetectable, and inversely (Figure 6 and Appendix A). HMGA2-expressing thyrocytes were predominantly localized in areas of the tumor where the follicular structure was altered, whereas thyroglobulin staining was detected in the colloid of follicles that had retained their structure. These results, obtained on three independent PTCs, highlighted the presence of considerable heterogeneity within PTCs and confirmed that HMGA2 protein expression was negatively correlated with thyroglobulin expression, in agreement with our mRNA expression data, but here at the cellular level.

### 2.7. Inhibition of HMGA2 by Short-Time Suramin Treatment Reduces Cell Invasion and Increases Differentiation Expression

Suramin, an antiparasitic drug, has previously been identified as a compound capable of strongly inhibiting HMGA2-DNA interactions [28], so we investigated the effects of this drug in TPC-1 and BCPAP cells. First, to estimate the cytotoxicity of suramin, we treated the cells with different concentrations of the drug (25, 50, 100, and 200 µM) for different times (1–4 days) and analyzed cell growth by counting the cells at days 1, 2, 3 and 4 (Figure 7). In both cell lines, we observed a significant slowdown in cell growth which was proportional to the concentration of suramin administered and the time of exposure, suggesting cell cytotoxicity [29,30]. Based on these results, to avoid the cytotoxic effect of suramin, we decided to use a concentration of suramin of 50 µM for 48 h in our further experiments.

Under these conditions, suramin had no impact on proliferation in TPC-1 and BCPAP cells, as shown by the number of EdU-labelled cells which did not vary after the treatment (Figure 8a). On the other hand, a significant reduction of the invasive capacity of both cell lines was observed following suramin treatment (Figure 8b), while no effect was observed on migration.

So far, suramin reproduced the effects previously observed following HMGA2 knockdown [11]. As mentioned above, inhibiting HMGA2 expression by siRNA did not modify the mRNA levels of several EMT markers in TPC-1 and BCPAP cells. Similarly, treating these cells by suramin did not induce any significant changes in SNAI1, SNAI2, ZEB1 and CDH1 mRNA expression (Figure 9), showing again that the drug was able to reproduce the effects of HMGA2 knockdown.

We then investigated whether suramin was able to reproduce the effects of HMGA2 knockdown on the expression of differentiation genes. We analyzed mRNA expression of TG, TPO and PAX8 in TPC-1 and BCPAP cells following a 48 h suramin treatment. A significant increase in TG and TPO mRNA expression was observed in both cell lines. Although not significant (*p* = 0.06), a trend towards increased expression of PAX8 mRNA was noticed (Figure 10). Overall, these data indicated that short-time inhibition of HMGA2 activity by suramin or knocking down HMGA2 expression by siRNA gave similar results, i.e., an inhibition of cell invasion and an induction of cell differentiation. Hence, HMGA2 may be a new molecular target for thyroid cancer, and our data suggest suramin as a potential new therapeutic approach.

## 3. Discussion

Over the past few decades, the number of detected thyroid cancer cases has significantly risen, likely due to the widespread adoption and improvement of ultrasound techniques. Traditional treatment options for thyroid cancer patients include total thyroidectomy, radioactive iodine therapy, and targeted therapies with tyrosine kinase inhibitors [31,32,33,34]. While many thyroid cancers respond favorably to radioiodine therapy, dedifferentiated cancers remain a significant challenge. These tumors have lost their ability to take up iodine, thus limiting the effectiveness of treatment. This dedifferentiation complicates clinical therapy management and highlights the necessity of identifying alternative therapeutic approaches to improve the prognosis of patients affected by these cancers. Despite recent therapeutic strategies focusing on redifferentiation to enhance the efficacity of RAI therapy, resistance mechanisms persist, necessitating the development of more effective treatment approaches for advanced thyroid cancers [35,36]. We have recently demonstrated that HMGA2 plays a prominent role in the aggressiveness of PTC cell lines by facilitating cell invasion [11], however the complete role of HMGA2 in thyroid tumorigenesis and the exact mechanism by which it exerts its effects remains poorly understood. In our previous work, we showed that HMGA2 expression is regulated by the MAPK pathway but not by the PI3K, or IGF1R signalling pathways [11]. This study aimed to elucidate more deeply the role of HMGA2 in PTC progression by investigating its potential involvement in dedifferentiation and EMT, both critical processes driving tumor aggressiveness and thyroid cancer progression. To address this, we first knocked down HMGA2 in two thyroid cancer derived cell lines: TPC-1 cells, derived from a RET/PTC1 positive PTC, and BCPAP cells, originating from a BRAF^V600E^ positive poorly differentiated PTC.

Our findings offer new insights into the function of HMGA2 in thyroid tumorigenesis. While HMGA2 does not appear to be involved in the induction of the EMT programme at transcriptional level and does not appear to participate in the TGFβ signalling pathway, we identified for the first time an association between HMGA2 and the process of cell dedifferentiation, since its specific knockdown in TPC-1 and BCPAP cells, following siRNA transfection, increased the mRNA expression of TG, TPO and PAX8, all related to thyroid function. Unfortunately, the basal expression of the NA^+^/I^−^ symporter (NIS), was undetectable in both TPC-1 and BCPAP cell lines, preventing the interpretability of the findings regarding the effect of HMGA2 on NIS expression. NIS is a key thyroid membrane glycoprotein which mediates the active transport of iodide from the blood stream into the thyroid tissue and the lack of expression or dysfunction of NIS is an important contributor to iodine resistant tumors [37]. The absence of NIS expression in our cells may result from the dedifferentiation acquired by most thyroid cancer cell lines derived from thyroid tumors, including TPC-1 and BCPAP cells, during their in vitro cell adaptation and evolution [38,39]. Consistent with our results, HMGA2 is related to the undifferentiated phenotype of immature leukemic cells and its expression level is inversely correlated with hepatic differentiation markers [40,41]. However, in adult stem cells, conflicting studies have depicted HMGA2 as playing a dual role in the differentiation process. Whereas HMGA2 overexpression blocks the differentiation of human mesenchymal stem cells into the osteogenic lineage by reducing RUNX2 expression [5], the study of O. Li et al. showed that HMGA2 was linked to mesenchymal differentiation [42], underlying that HMGA2 role in cell differentiation might depend on the specific cellular context.

In addition, our research demonstrated that the inhibition of the MAPK pathway in TPC-1 and BCPAP cells, known to decrease both HMGA2 mRNA and protein expression [11], reproduced the increase in TG, TPO and PAX8 mRNA expression. Restoration of thyroid differentiation following inhibition of the MAPK pathway has already been reported in multiple in vitro studies [43,44,45,46]. In transgenic mice with BRAF^V600E^ protein activation, suppression of the MAPK signaling pathway using BRAF and MEK inhibitors also showed restoration of differentiation, as well as radioactive iodine uptake [47]. Consistent with this, the BRAF^V600E^ mutation is known to upregulate epigenetic pathways that silence the expression of the sodium/iodide symporter NIS [48]. More generally, the expression of differentiation markers (NIS, TG, TPO, TSHR…) is lower in BRAF^V600E^ mutated PTC than in non BRAF mutated PTC [49,50]. When comparing HMGA2 expression in BRAF mutated and non-mutated PTC from the TCGA, it appeared that HMGA2 mRNA levels are higher in the BRAF^V600E^ positive samples and also in higher-stage tumors [11], supporting the fact that HMGA2 may contribute to the more dedifferentiated and aggressive phenotype of these tumors. On the other hand, activation of the cAMP-dependent signalling pathway, inducing proliferation and differentiation of thyrocytes [51], by forskolin led to a decrease in HMGA2 expression and an increase in the expression of thyroid differentiation markers.

Overall, our findings suggest that HMGA2 plays a significant role in the dedifferentiation of PTC cells in response to the stimulation of the MAPK cascade or to the inhibition of the cAMP-dependent signalling pathway and subsequently, contributes to the aggressiveness of thyroid tumors. Interestingly, HMGA2 was found to enhance the activation of several crucial pathways, including the MAPK, Wnt/β-catenin, and mTOR signaling pathways, suggesting a potential positive feedback loop [4,14,52,53]. However, further investigations are required to fully elucidate the relationship between HMGA2 and these signaling pathways. Our findings are not restricted to thyroid cancer since HMGA2 has several times been identified as correlating with tumor aggressiveness in other cancers such as breast cancer [54], tongue squamous cell carcinoma [55] and ovarian cancer [56,57], highlighting the common oncogenic role of HMGA2 in tumor progression.

Although cell lines are crucial to elucidate mechanisms of tumorigenesis, they are only partial models of the in vivo corresponding tumors, and this constitutes a limitation of our study. The TPC-1 and BCPAP cell lines are widely used in research. While not representing the full spectrum of PTC, they reflect the most common cases. Even if these cell lines have retained their initial genetic alteration (RET/PTC1 rearrangement or BRAF mutation), they do not recapitulate all the characteristics of in vivo PTC: in addition to loss of tissue heterogeneity and microenvironment differences, the cell lines have evolved and diverged during their in vitro adaptation [38,39].

The use of mouse models of thyroid cancer allows to confirm and to further investigate the role of HMGA2 in cell invasion and the relationship between HMGA2 and thyroid differentiation. RET/PTC3 transgenic mice, in which the expression of the RET/PTC3 oncogene is targeted to the thyroid, were used in this study. The RET/PTC3 rearrangement encodes a fusion protein comprising the RET tyrosine kinase domain and the 5′ terminal region of the ELE1 gene. The expression of this oncoprotein induces the constitutive activation of the MAPK and PI3K signaling pathways and leads to the development of solid variant of papillary thyroid carcinoma, representing a suitable model of human PTC [17,18,25]. However, RET/PTC3 transgenic mice are partial and transient models of human PTC. They present several human PTC characteristics, but similarities with human PTC are incomplete, as they do not concern the overall gene expression and are not conserved in old animals. Nevertheless, they provide a good model for several major properties of human PTC in 2-month-old mice [18]. At two months of age, thyroid differentiation genes and transcription factors were expressed, consistent with findings from Burniat and colleagues [18]. HMGA2 mRNA levels were increased and negatively correlated with TG and NIS mRNA levels. Similarly, by six months of age, HMGA2 mRNA expression was increased and showed negative correlations with TG and PAX8 mRNA expression. Although NIS mRNA expression was significantly downregulated, we only noticed a trend towards a negative correlation between HMGA2 and NIS mRNA levels. We also found increased TSHR and NKX2.1 mRNA levels in the 6-month-old mice, which contrasts with studies in human thyroid cancers reporting that the expression of thyroid differentiation genes is unchanged, decreased or lost [58,59]. This discrepancy suggests that this mouse model is not ideal for studying tumor progression after the age of two months, as already mentioned earlier [18] and does not accurately represent a model of either poorly differentiated thyroid cancer (PDTC) or anaplastic thyroid cancer (ATC). Based on our results, 2-month-old RET/PTC3 mice represent the best model to further explore the molecular function of HMGA2 in vivo. In the future, conducting similar experiments in BRAF^V600E^ transgenic mice would help us to extend our findings.

The clinical relevance of our results was then evaluated in human PTC. By analyzing in silico data from TCGA (502 PTC, 59 normal tissues), we found a negative correlation between HMGA2 and TG, TPO, NIS, or PAX8 mRNA expressions. Consistently, in independent PTC samples, we observed a significant negative correlation between HMGA2 and TPO mRNA expressions, which aligns with our in vitro findings. Similar trends were observed for TG, NIS and PAX8, although not statistically significant because of the small number of samples available. We further validated the increased levels of HMGA2 and decreased levels of TG at protein level by performing immunohistological analyses in three PTC. Our data revealed significant cell heterogeneity in PTC, showing different states of cancer cell differentiation. Specifically, we observed areas where cells expressed HMGA2 but not TG, and conversely, areas where cells expressed TG but not HMGA2. HMGA2+/TG− cells were observed in less differentiated areas, showing altered follicular organization, while HMGA2−/TG+ cells were detected in regions still presenting some follicular structure. The negative correlation between HMGA2 and TG expression was thus also present at protein level.

Due to its low expression in adult normal tissues, HMGA2 emerges as a promising therapeutic target, with potential minimal side effects. Suramin, an anti-trypanosomal drug used to treat African sleeping sickness and river blindness, has recently been identified as a potent inhibitor of HMGA2-DNA interactions [28,60]. Suramin has been tested in various clinical trials for cancer therapy due to its diverse properties. It is known for its ability to bind and displace growth factors from their receptors, which are often overexpressed in tumors. It has been investigated to treat several cancer types including prostate cancer [61,62], non-small cell lung cancer [63], breast cancer [63], bladder cancer [64], and brain tumors [65]. Although suramin binds to various targets, the study of Su and colleagues recently showed that suramin was a potent inhibitor for HMGA2-DNA interactions [28]. We explored the functional role of suramin in vitro in TPC-1 and BCPAP cells and showed that short time exposure to this drug has similar effects to siRNA inhibition of HMGA2 expression, i.e., no impact on cell proliferation or transcription of EMT-related genes, inhibition of cell invasion and re-expression of thyroid function related-genes. Our current findings are promising for the potential therapeutic utility of suramin in the treatment of thyroid cancer, which should be compared to established redifferentiation therapies. Suramin has shown even more promising results when used in combination with various chemotherapeutic agents, as demonstrated by numerous studies on mouse models and clinical trials, for example in patients with hormone-refractory prostate cancer [61,66,67], NSCLC [68], in a post-surgery and post-radiotherapy context like a metastasis preventing strategy [69], or in the treatment of corneal neovascularization [70]. It would therefore be very interesting to test the efficacy of suramin in combination with trametinib, dabrafenib or other kinase inhibitors, commonly used in thyroid cancer therapy, and our RET/PTC3 mouse model is perfectly suited to do that in the future.

To conclude, the characterisation of HMGA2 in thyroid tumorigenesis suggests its potential involvement in cell dedifferentiation and tumor progression of thyroid cancers. Overexpression of HMGA2 could be one of the mechanisms behind PTC dedifferentiation and resistance to radioactive iodine treatment. However, the precise mechanisms by which HMGA2 is involved in dedifferentiation are still unknown and need to be further investigated. Inhibiting HMGA2 activity using suramin showed promising results in vitro with few non-specific effects, namely a significant anti-tumor effect by inhibiting cell invasion and reinducing differentiation, mimicking the effects resulting from knocking down HMGA2 by siRNA. So, HMGA2 may be a new molecular target for thyroid cancer, and our data suggest suramin as a potential new therapeutic approach.

## 4. Materials and Methods

### 4.1. Tissue Collection

Frozen human thyroid tissues were obtained from the J. Bordet Institute. All samples were stained by hematoxylin and eosin and their pathological status were confirmed by an anatomopathologist from the J. Bordet Institute. Protocols have been approved by the ethics committees of the institutions (protocol 1978). Written informed consent was obtained from all participants involved in the study. Clinical information is given in Appendix A.

### 4.2. Cell Lines and Treatments

The BCPAP cell line is derived from a BRAF^V600E^ positive poorly differentiated PTC and was received from Prof. G. Brabant (Department of Internal Medicine I, Lübeck, Germany). The TPC1 cell line is derived from a RET/PTC1 positive PTC and was obtained from Dr. M. Mareel (University of Ghent, Belgium). The STR profile of both cell lines was performed to ensure their purity and identity. All cell lines were maintained into RPMI1640 media (Gibco/Thermo Fisher Scientific, Waltham, MA, USA) supplemented with 10% of fetal bovine serum (FBS) at 37 °C in a humidified atmosphere containing 5% of CO_2_. Cells were treated with a MEK inhibitor (trametinib, 10 nM, #GSK1120212; S2673, SelleckChem, Houston, TX, USA) for 24 h. Treatments with suramin (#574625, Merck, Rahway, NJ, USA) were performed in gradient concentrations of 25, 50, 100 and 200 µM, administered for different times (1–4 days). Treatments with TGFβ (#100-21C, Thermo Fischer Scientific, Waltham, MA, USA) were tested in a gradient of 1, 10, 20 and 100 ng/mL for different times, before cells were harvested for RNA or protein extraction. The effectiveness of the MEK inhibitor treatment was confirmed by western blot analysis [11].

### 4.3. Transfection Experiments

Cells were transiently transfected with siRNA silencer select HMGA2 (#4392420) or negative control (#4390846) (Thermo Fisher Scientific, Waltham, MA, USA), at a concentration of 20 nM. The transfection was performed using Lipofectamine RNAiMAX reagent (Invitrogen/Thermo Fisher Scientific, Waltham, MA, USA) according to the manufacturer’s protocol. The silencing of HMGA2 was verified after each transfection by RT-qPCR analysis.

### 4.4. RNA Purification and Real-Time PCR Analysis

Total RNA was isolated from cells using the miRNeasy Mini Kit (Qiagen, Hilden, Germany) according to the manufacturer’s instructions. The Superscript III reverse transcription kit (Invitrogen/Thermo Fisher Scientific, Waltham, MA, USA) was used for reverse transcription. PCR amplification was performed using KAPA SYBR FAST (Kapa Biosystems, Wilmington, MA, USA) for mRNA quantification on ABI 7500 detection system (Biorad, Hercules, CA, USA). NEDD8 (NEDD8 Ubiquitin Like Modifier) and TTC1 (Tetratricopeptide Repeat Domain 1) were used as internal normalizers for human samples, and TBP (TATA binding protein) and HPRT (Hypoxanthine Phosphoribosyltransferase 1) were used for mouse samples. The relative expression of each gene was calculated and normalized using the 2^−ΔCt^ method [71]. The primer sequences used for mRNA amplification are listed in Appendix A.

### 4.5. Cell Migration and Invasion Assays

Cell migration and invasion were analyzed by using 24-well Transwell chamber assays (8 µm pore size) (VWR 734-0038 and VWR 734-1047, Corning, Corning, NY, USA), as described in the manufacturer’s instructions. TPC-1 and BCPAP cells were seeded into 6 well-plates and transfected as described above. After 48 h, the medium was replaced by fresh medium in the absence of serum. The cells were harvested 3 days after transfection with a solution of PBS/EDTA (5 nM)/EGTA (5 nM) and 4 × 10^4^ cells per well were seeded in the upper chamber containing 500 µL of medium without serum whereas the lower chamber contained 750 µL of medium with FBS as attractant. After 20 h of incubation, the cells of the upper membrane were gently removed using a cotton swab and the cells that have passed through were fixed and stained with azure and xanthene dyes (Polysciences, Inc., Warrington, PA, USA). Cells were counted in 5 random fields per well using a microscope (ZOE TM fluorescent Cell imager, Biorad, Hercules, CA, USA). For the invasion assay, the procedures were the same except that the inserts were coated with matrigel. The percentage of invasion was calculated using the following formula: (invasive cell count/migrative cell count) × 100.

### 4.6. Proliferation Assay

15 × 10^3^ cells were seeded into 6-well plates and proliferation was analyzed 72 h after transfection by using the Click-iT Plus EdU Proliferation Kit (Thermo Fisher Scientific, Waltham, MA, USA) according to the manufacturer’s instructions. Briefly, the cells were incubated for 6 h with 5′-ethynyl-2′-deoxyuridine at 37 °C in the dark and EdU incorporation was analyzed by flow cytometry on a BD LSRFortessa cell analyzer (FACS).

### 4.7. Western Blotting

For Western Blotting, lysis and extraction of total proteins from tissues and cells were performed on ice using LAEMMLI buffer supplemented with protease and phosphatase inhibitors. Protein concentration was determined using Ionic Detergent Compatibility Reagent (IDCR) for Pierce (Thermo Fisher Scientific, Waltham, MA, USA). Denatured proteins (30 µg) were separated by 8% sodium dodecyl sulphate-polyacrylamide gel electrophoresis (SDS-PAGE) and then transferred to nitrocellulose membranes. Primary antibodies were incubated at 4 °C overnight and secondary HRP-conjugated goat anti-bodies at room temperature for 1 h. Antibodies against P-SMAD2 were purchased from Cell Signaling Technology (#3108, Danvers, MA, USA). Protein bands were visualized by chemiluminescence with ECL Prime Western Detection Reagent (PerkinElmer, Waltham, MA, USA). Protein levels were quantified and normalized to vinculin protein levels in each sample by densitometry analysis using ImageJ 1.54d software.

### 4.8. Immunofluorescence

For immunofluorescence staining, tissue sections (7 µm of thickness) were fixed with 4% paraformaldehyde during 10 min and then washed two times with PBS. Subsequently, sections were blocked with a permeabilized and blocking solution containing 1% BSA 0.2% Triton X-100 5% NHS PBS for 1 h and then incubated with polyclonal rabbit antibodies against HMGA2 (#8179S, Cell Signaling Technology, 1:200), TG (A0251, Dako, 1:3000) or cytokeratin 8 (Merck, #TROMA-I, 1:500) in a humidified chamber at 4 °C overnight. After three washes with PBS-Tween 0.05%, sections were incubated with Alexa Fluor 488 conjugated goat anti rabbit IgG (Invitrogen, #A21206), Alexa Fluor 546 conjugated goat anti rat IgG (Invitrogen, #A11081) and DAPI (1:30,000) for 1 h and then washed 3 times with PBS-Tween 0.05% and PBS. The images were scanned with a ZEISS Axio microscope and protein levels were quantified using the QuPATH v0.5.1 software.

### 4.9. RET/PTC3 Mice Model

RET/PTC3 transgenic mice [17] were provided by Dr. Decaussin-Petrucci (Department of Pathology, CHU Lyon, France). At 2 and 6 months, mice were sacrificed for thyroid removal. Dissected tissues were immediately placed on ice, snap-frozen in liquid nitrogen and stored at −80 °C until RNA or protein processing. All animal procedures were reviewed and approved by the Animal Care and Use Committee of the university (#CEBEA-IBMM-2020-25-86).

### 4.10. Statistical Analyses

Statistical analyses were performed using Prism GraphPad 6.0. Data distribution was analyzed by Shapiro-Wilk normality test. The data collected in this work had a normal distribution (*p*-value of Shapiro-Wilk normality was <0.05). The t-test was used for data analysis between 2 groups and when ≥3 groups were compared, the ANOVA test was applied. Quantitative data are represented as mean and standard deviation. All experiments were replicated independently at least 3 times. A *p*-value less than 0.05 was considered statistically significant.

## Figures and Tables

**Figure 1 ijms-26-01643-f001:**
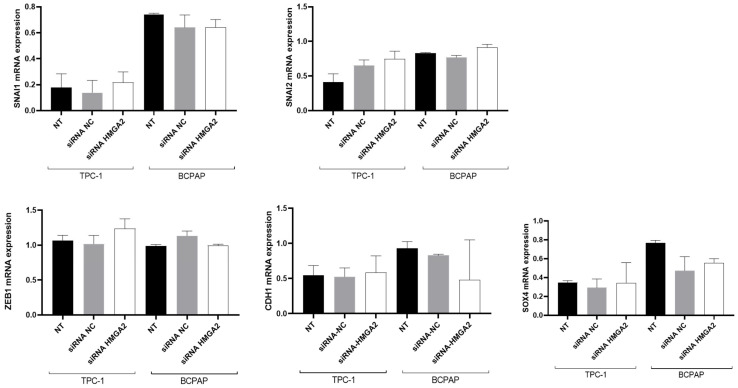
HMGA2 does not modulate the mRNA expression of SNAI1, SNAI2, ZEB1, CDH1 and SOX4 in TPC-1 and BCPAP cells. Analysis of EMT markers gene expression by RT-qPCR in non-transfected TPC-1 and BCPAP cells (NT) and three days after transfection with a siRNA against HMGA2 (siRNA HMGA2) or a control siRNA (siRNA NC) (*n* = 3).

**Figure 2 ijms-26-01643-f002:**
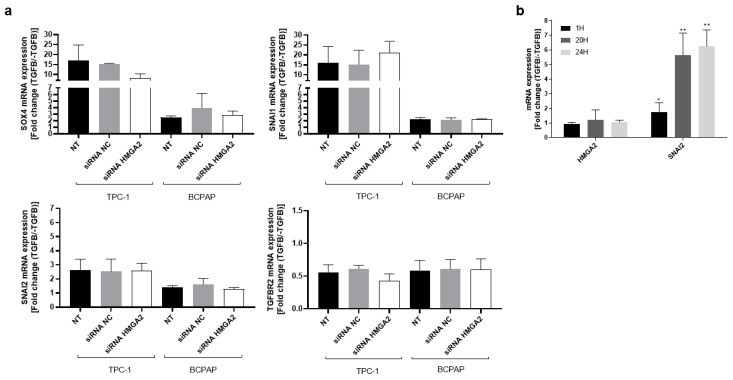
HMGA2 does not significantly modulate the mRNA expression of SOX4, SNAI1, SNAI2 and TGFBR2 in response to TGFβ in TPC-1 and B-CPAP cells. (**a**) Analysis of the mRNA expression of SOX4, SNAI1, SNAI2 and TGFBR2 by RT-qPCR in TPC-1 and BCPAP cells three days after transfection with a siRNA against HMGA2 (siRNA HMGA2) or a control siRNA (siRNA NC) and treated with TGFβ for 24 h at 10 ng/mL (NT: non transfected cells). Expression ratio for each gene was performed between TGFβ-treated and untreated TPC-1 or BCPAP cells (Fold change (TGFβ/-TGFβ) (*n* = 3)). (**b**) Analysis of the mRNA expression of HMGA2 and SNAI2 by RT-qPCR in TPC-1 cells after 1 h, 20 h and 24 h TGFβ treatment (10 ng/mL). An expression ratio for each gene was performed between TGFβ-treated and untreated TPC-1 cells (Fold change (TGFβ/-TGFβ) (*n* = 3)). * *p* < 0.05 and ** *p* < 0.01 vs. TGFβ-untreated TPC-1 cells.

**Figure 3 ijms-26-01643-f003:**
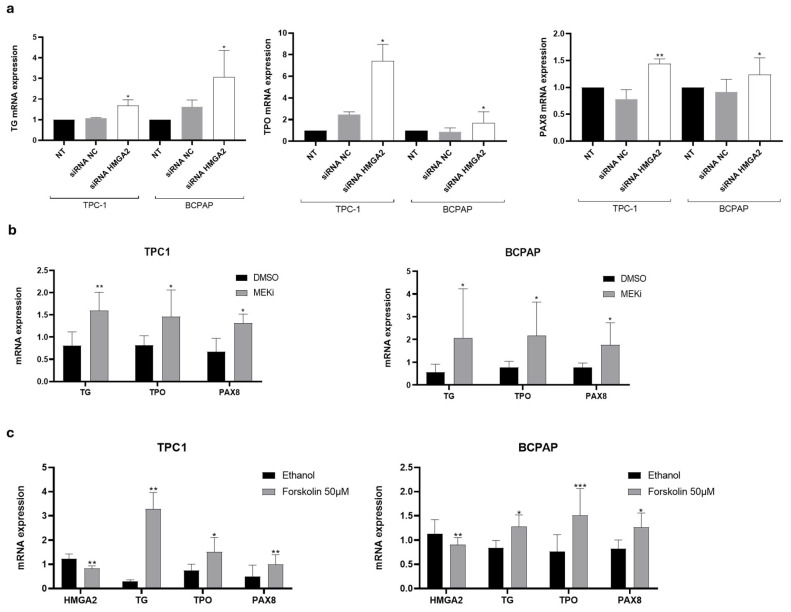
Inhibition of HMGA2 significantly increases the mRNA expression of TG, TPO and PAX8 in TPC-1 and BCPAP cells. (**a**) Analysis of mRNA expression of TG, TPO and PAX8 by RT-qPCR in non-transfected TPC-1 and BCPAP cells (NT) and three days after transfection with a control siRNA (siRNA NC) or a siRNA directed against HMGA2 (siRNA HMGA2) (*n* = 4). * *p* < 0.05, ** *p* < 0.01, vs. siRNA NC (**b**) Analysis of mRNA expression in TPC-1 and BCPAP cells treated with trametinib (MEKi 10 µM) or DMSO for 24 h. * *p* < 0.05; ** *p* < 0.01, vs. DMSO (*n* = 4) (**c**) Analysis of mRNA expression in TPC-1 and BCPAP cells treated with forskolin (50 µM) or ethanol for 48 h. * *p* < 0.05; ** *p* < 0.01, *** *p* < 0.001 vs. ethanol (*n* = 6).

**Figure 4 ijms-26-01643-f004:**
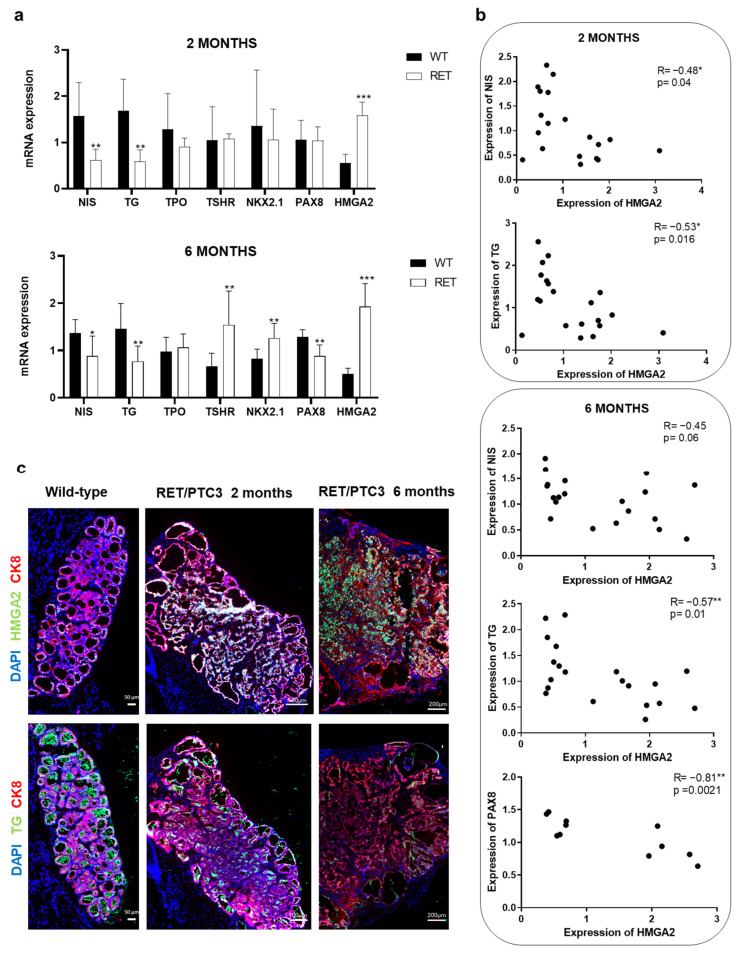
HMGA2 is overexpressed while thyroid differentiation genes are downregulated in thyroids from RET/PTC3 mice. (**a**) RT-qPCR analysis of NIS, TG, TPO, TSHR, NKX2.1, PAX8 and HMGA2 mRNA expression in 2- and 6-month-old RET/PTC3 mice thyroids (RET) (*n* = 10) and in a pool of normal thyroids (WT) (*n* = 10).* *p* < 0.05, ** *p* < 0.01 and *** *p* < 0.001 vs. WT. (**b**) Correlation analyses between HMGA2 and NIS (*n* = 19) or TG (*n* = 20) mRNA expression in thyroids from two-month-old mice and between HMGA2 and NIS (*n* = 20), TG (*n* = 20) or PAX8 (*n* = 11) mRNA expression in thyroids from 6-month-old mice. R = Pearson correlation coefficient. * *p* < 0.05 and ** *p* < 0.01. (**c**) Analysis of protein expression by immunostaining of HMGA2 (green) and TG (green) in mice thyroids at 2 and 6 months. Epithelial cells were labeled with anti-CK8 antibody (red) and nuclei with DAPI (blue).

**Figure 5 ijms-26-01643-f005:**
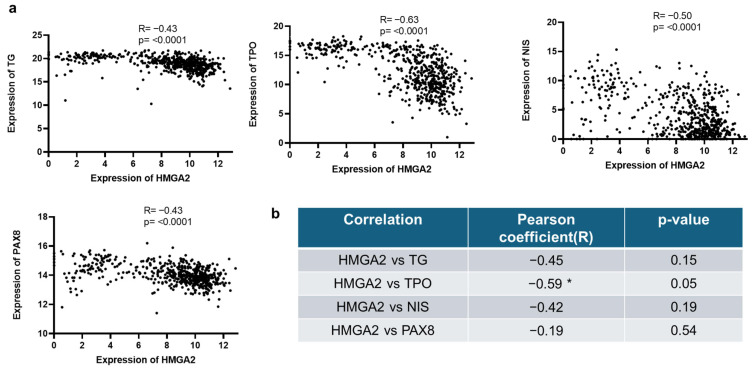
HMGA2 mRNA expression is inversely correlated to TG, TPO, NIS and PAX8 mRNA expression in human PTC. Correlation analysis was performed between HMGA2 and TG, TPO, NIS or PAX8 mRNA expression (**a**) in human PTC based on TCGA RNA-seq data (502 PTC, 59 Normal tissues) and (**b**) in 10 (TG, NIS) or 11 (TPO, PAX8) independent human PTC samples following RT-qPCR analyses. R = Pearson correlation coefficient. * *p* < 0.05.

**Figure 6 ijms-26-01643-f006:**
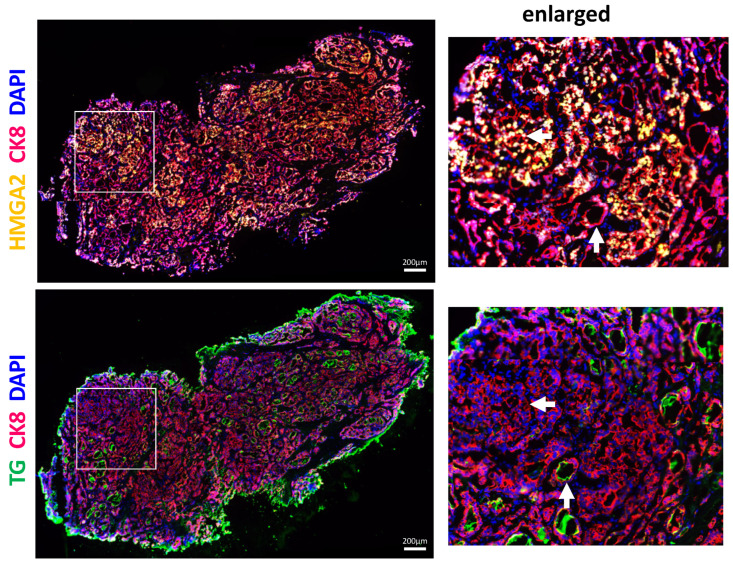
HMGA2 protein expression is negatively correlated with TG protein expression in human PTC at cellular resolution level. Immunostaining of HMGA2 (yellow) and TG (green) on two adjacent sections of a representative PTC reveals the presence of HMGA2+/TG− (horizontal arrows on the enlarged area) and HMGA2−/TG+ cells (vertical arrows on the enlarged area). Epithelial cells were labeled with anti-CK8 antibody (red) and nuclei with DAPI (blue).

**Figure 7 ijms-26-01643-f007:**
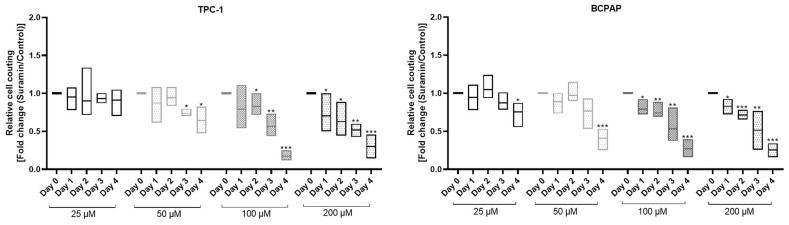
Dose and time course effects of suramin in TPC-1 and BCPAP cells. Cells were treated with increasing concentrations of suramin (25, 50, 100 and 200 μM) for 4 days and counted (control: untreated cells) (Fold change (Suramin/Control)) * *p* < 0.05; ** *p* < 0.01, *** *p* < 0.001 vs. untreated cells (*n* = 4).

**Figure 8 ijms-26-01643-f008:**
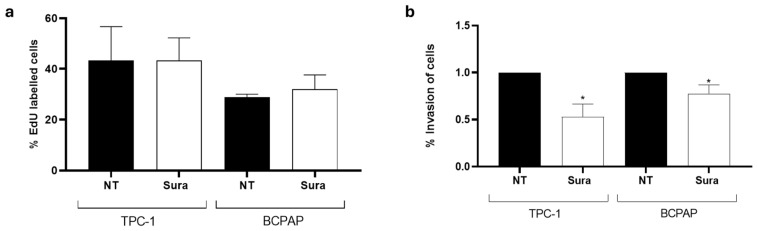
Short time suramin treatment has no effect on proliferation but inhibits cell invasion. (**a**) EdU incorporation was analyzed by flow cytometry in TPC-1 and BCPAP cells, untreated (NT) and treated with suramin for 48 h (Sura, 50 μM) (*n* = 4). (**b**) The percentage of invasion was analyzed in TPC-1 and BCPAP cells, untreated (NT) and treated with suramin (Sura, 50 μM) for 48 h, by counting the cell in 5 random fields. Invasion was defined as follows: (mean number of invading cells/mean number of migrating cells) ∗ 100 (*n* = 3). * *p* < 0.05.

**Figure 9 ijms-26-01643-f009:**
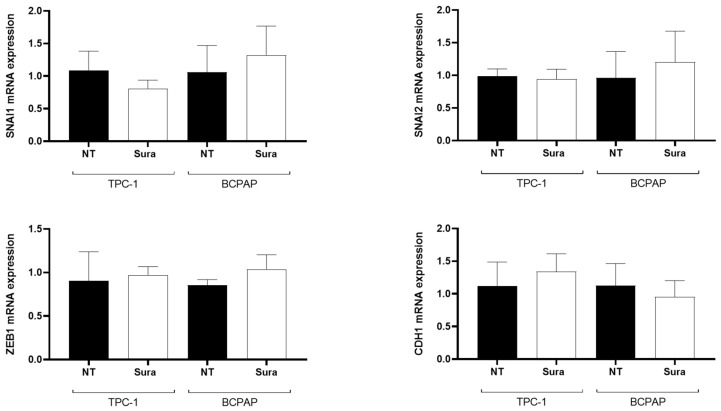
Short time suramin treatment does not modulate the expression of EMT related genes.dAnalysis of the mRNA expression of SNAI1, SNAI2, ZEB1 and CDH1 by RT-qPCR in TPC-1 and BCPAP cells, untreated (NT) and treated with suramin (Sura, 50 µM) for 48 h (*n* = 3).

**Figure 10 ijms-26-01643-f010:**
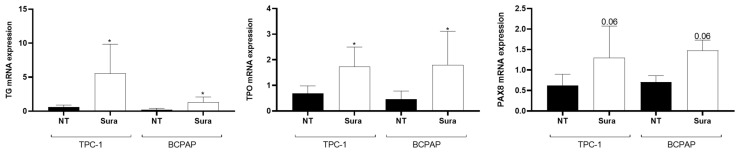
Suramin increases the mRNA expression of thyroid differentiation genes such as TG (*n* = 4), TPO (*n* = 7) and PAX8 (*n* = 5) in TPC-1 and BCPAP cells. Analysis of TG, TPO and PAX8 mRNA expression by RT-qPCR in TPC-1 and BCPAP cells, untreated (NT) and treated with suramin (50 µM) (Sura) for 48 h. * *p* < 0.05.

## Data Availability

The original contributions presented in the study are included in the article/Appendix A, further inquiries can be directed to the corresponding author.

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
