# Peer review of "HMGA2 Overexpression in Papillary Thyroid Cancer Promotes Thyroid Cell Dedifferentiation and Invasion, and These Effects Are Counteracted by Suramin"

_ijms, 2025, doi:10.3390/ijms26041643_

Round 1

Reviewer 1 Report

Comments and Suggestions for Authors

Dear authors,

Your manuscript "HMGA2 overexpression in papillary thyroid cancer promotes thyroid cell dedifferentiation and invasion, and these effects are counteracted by suramin," ijms-3383629, which discovers the role of HMGA2 in the development of aggressiveness in papillary thyroid carcinoma, as well as its potential role as a target in thyroid carcinoma therapy, is well-written and well-presented; it is easy to follow and contains interesting results and observations. Therefore, I recommend publishing your manuscript after making a few minor corrections:

1.        Due to the correctness and clarity of the data presented in a couple of places in the manuscript the words that better determine tested factor should be added:

a.        The sentence in line 79: “… were analyzed in a transgenic mouse model of PTC as well as in human PTC” should be complemented with the model of human PTC or analysis type. As I could understand, TCGA analysis was performed. Therefore, I suggest you add something like “…the data gained by TCGA analysis in human PTC”.

b.       Similarly, in the next line (line 80) “Finally, we explored the therapeutic potential of suramin for treating thyroid cancer” the model system should be added (cell lines or cell culture).

c.        In the title of the 2.3 subsection, the word “gene” should be added: “Inhibition of HMGA2 gene induces the expression of thyroid differentiation genes in PTC cell lines”.

d.       In line 162: “…in the expression of TG, TPO and PAX8 was observed after HMGA2 knockdown”, the word “mRNA” should be added: “… increase in the expression of TG, TPO and PAX8 mRNAs was observed after HMGA2 knockdown”.

e.        In line 163: “Basal expression of NIS was undetectable…” the word “mRNA” is missing. I suggest: “Basal expression of NIS mRNA was undetectable…”

2.        In this work the increase in mRNA levels was demonstrated. It would be beneficial if protein levels could be tested, evaluated, and presented, too.

3.        The results gained on diverse model subtypes used in divergent parts of the manuscript should be discussed in the discussion section too, not just mentioned in the Material and Methods section. The RET/PTC3 model system of mice develops a solid variant of PTC. TPC1 cell lines originate from RET/PTC1 positive PTC, BCPAP originates from BRAFV600E positive poorly differentiated PTC. And for the analysis of the data extracted from the TCGA database, the whole PTC sample was included. In addition, some papillary thyroid carcinoma or follicular thyroid carcinoma (PTC or FTC) cell lines (i.e., TPC-1) might have partially lost their original DNA synthesis/replication regulation mechanisms during their in vitro cell adaptation/evolution (10.3389/fendo.2012.00133).

4.        The conclusion stated in lines 321-323 is too ambitious: “HMGA2 therefore appears to be a new molecular target for the treatment of thyroid cancer, and our data suggest the use of suramin as a new therapeutic approach.” Some additional experiments and clinical studies should be employed for such closing. Rewrite this sentence as: "HMGA2 may be a new molecular target for thyroid cancer, and our data suggest suramin as a potential new therapeutic approach..." Similarly, the conclusion indicated in lines 475–476 should be adapted.

5.        In line 514, it is written that the relative expression of each gene was calculated and normalized using the 2-ΔΔCt method [69]. Firstly, please add the name of the gene used as an endogen control and its primers’ sequences used for the amplification. Secondly, correct this part, as it seems that the 2-ΔCt method was applied, not the 2-ΔΔCt. For the application of the 2-ΔΔCt method, besides endogen control (reference gene), some control tissue is needed (e.g., healthy tissue) as a calibrator, and in this work, there was no control tissue. In other words, in your work the relative expression of the genes was measured, not their fold change. Therefore, the method should be corrected.

6.        In line 576, it is written that “Data distribution was analyzed by the Shapiro-Wilk normality test”. If so, add the line of the results of the test application, e.g., “the data gained in this work had a normal distribution (Shapiro-Wilk normality test….)”

7.        Please rewrite the sentence in line 577: “The Friedman Anova test was used for multiple data analysis…” as in your work, no multiple comparison was applied, but the difference between more than 2 groups of samples was tested. I suggest you revise it to “when ≥3 groups were compared, the ANOVA test was applied.”

Reviewer 2 Report

Comments and Suggestions for Authors

A manuscript submitted by Branteghem and colleagues highlights the role of HMGA2 in papillary thyroid cancer (PTC), showing that its overexpression promotes thyroid cell dedifferentiation and invasion, while its inhibition—either through siRNA knockdown or treatment with suramin—restores differentiation markers and reduces invasiveness. The findings suggest that HMGA2 could be a potential therapeutic target for PTC, with suramin presenting a promising treatment strategy to mitigate tumor aggressiveness.

The manuscript is compelling and suitable for publication; however, the authors should clarify a few minor points:

  • In the siRNA knockdown experiments, how was off-target gene silencing ruled out to confirm specificity for HMGA2?
  • The study utilizes TPC-1 and BCPAP cell lines. Do these models adequately represent the full spectrum of PTC subtypes, particularly concerning genetic heterogeneity?
  • Were any additional molecular pathways (e.g., PI3K/AKT, Wnt/β-catenin) investigated in relation to HMGA2 beyond MAPK and TGF-β signaling?
  • What are the limitations of using RET/PTC3 transgenic mice as a model for human PTC, and how do these findings apply to other genetic mutations such as BRAFV600E?
  • The study proposes suramin as a potential therapeutic agent. How does its efficacy compare to established redifferentiation therapies, such as MEK inhibitors?

Round 2

Reviewer 2 Report

Comments and Suggestions for Authors

The authors have addressed all my questions